# Effects of Transcranial Direct Current Stimulation over the Supplementary Motor Area Combined with Walking on the Intramuscular Coherence of the Tibialis Anterior in a Subacute Post-Stroke Patient: A Single-Case Study

**DOI:** 10.3390/brainsci12050540

**Published:** 2022-04-23

**Authors:** Naruhito Hasui, Naomichi Mizuta, Junji Taguchi, Tomoki Nakatani, Shu Morioka

**Affiliations:** 1Department of Therapy, Takarazuka Rehabilitation Hospital, Medical Corporation SHOWAKAI, 22-2 Tsuru-no-so, Takarazuka 665-0833, Japan; peace.pt1028@gmail.com (N.M.); taguchi@takara-reha.com (J.T.); ryouhoushi@gmail.com (T.N.); 2Department of Neurorehabilitation, Graduate School of Health Sciences, Kio University, 4-2-2 Umaminaka, Koryo, Kitakatsuragi-gun, Nara 635-0832, Japan; s.morioka@kio.ac.jp; 3Neurorehabilitation Research Center, Kio University, 4-2-2 Umaminaka, Koryo, Kitakatsuragi-gun, Nara 635-0832, Japan; 4Department of Rehabilitation, Faculty of Health Sciences, Nihon Fukushi University, 26-2 Higashihaemi-cho, Handa 475-0012, Aichi, Japan

**Keywords:** post-stroke, walking, transcranial direct current stimulation, supplementary motor area, intramuscular coherence, gait variability

## Abstract

Motor recovery is related to the corticospinal tract (CST) lesion in post-stroke patients. The CST originating from the supplementary motor area (SMA) affects the recovery of impaired motor function. We confirmed the effects of transcranial direct current stimulation (tDCS) over the SMA combined with walk training on CST excitability. This study involved a stroke patient with severe sensorimotor deficits and a retrospective AB design. Walk training was conducted only in phase A. Phase B consisted of anodal tDCS (1.5 mA) combined with walk training. Walking speed, stride time variability (STV; reflecting gait stability), and beta-band intramuscular coherence—derived from the paired tibialis anterior on the paretic side (reflecting CST excitability)—were measured. STV quantified the coefficient of variation in stride time using accelerometers. Intramuscular coherence during the early stance phase noticeably increased in phase B compared with phase A. Intramuscular coherence in both the stance and swing phases was reduced at follow-up. Walking speed showed no change, while STV was noticeably decreased in phase B compared with phase A. These results suggest that tDCS over the SMA during walking improves gait stability by enhancing CST excitability in the early stance phase.

## 1. Introduction

Walking in humans is primarily controlled by the brainstem and spinal cord mechanisms [1,2]. A recent study reported that cortical activity is also involved in human overground walking [3]. In particular, the activity of the corticospinal tract (CST) during walking is phase-dependent and is enhanced during the early stance and swing phases [4]. CST excitability in the early stance phase has been shown to be a stable strategy to prepare for disturbances after heel strikes [5]. Therefore, walking is considered to be accomplished not only via subcortical control but also through cortical control. The plasticity of the damaged CST is a major factor affecting motor recovery in post-stroke patients [6,7]. Post-stroke patients with CST damage typically exhibit clinical symptoms that include slow walking speed, increased gait asymmetry, and poor dynamic balance [8,9,10]. In particular, motor paralysis after stroke is strongly affected by walking impairments [11]. Moreover, in post-stroke patients with severe CST damage, the CST originating from the supplementary motor area (SMA), as well as the primary motor area (M1), impacts motor outcomes [12,13]. This recovery indicates a compensatory mechanism in which M1 provides most of the CST fibers; however, the CST originating from the SMA contributes to contralateral movements when the CST originating from M1 is severely damaged [12,13]. As the type of motor recovery differs depending on the severity of the CST injury, we consider it necessary to increase the activity of M1 or SMA according to the recovery type. Therefore, the effects of SMA on the CST excitability of the lower limb muscles require experimental manipulation of SMA activity, which should be clarified by a longitudinal study design.

Transcranial direct current stimulation (tDCS) is a non-invasive method for enhancing cortical activity [14]. In fact, corticomuscular and intramuscular coherence is increased by using tDCS for M1 [15,16]. M1-targeted tDCS in post-stroke patients improves mobility and paretic lower limb motor function [17]; however, the effects of tDCS on lower limb muscle activity and motor evoked potentials (MEP) during walking have been shown to be inconsistent [18,19,20]. We considered this variability to be reflective of differences in the type of recovery depending on the severity of the CST injury because the patient in these previous studies had a drop foot (−1.7°) during the swing phase and motor paralysis of the lower limb (Fugl–Meyer Assessment for lower extremities: 23.4–30.9 points) [18,19,20]. Additionally, walk training with tDCS targeting SMA improves walking speed and performance in the Timed Up and Go test [21]; however, its effect on CST activity during walking is unclear.

The purpose of this study was to examine the effects of tDCS over the SMA combined with walk training on CST excitability during walking in a subacute post-stroke patient. The participant in this study had severe motor paralysis of the lower limb caused by a CST lesion. We hypothesized that walk training with tDCS over the SMA would increase CST excitability in patients with severe motor paralysis. The significance of this study was to determine the effects of tDCS on the SMA in a subacute post-stroke patient with reduced CST activation during walking. These results can contribute to the clinical application of neuromodulation for specific types of post-stroke recovery.

## 2. Materials and Methods

### 2.1. Participant

The participant (68-year-old man) was a stroke patient with a lesion in the left caudate nucleus, internal capsule, and corona radiata due to middle cerebral artery infarction who had severe sensorimotor deficits (Figure 1). Regarding pre-stroke activities, the patient was a member of the town council and frequently went outdoors. After treatment at an acute care hospital, the patient was admitted to the Takarazuka Rehabilitation Hospital for intensive rehabilitation 15 days after the stroke onset. Unfortunately, this patient developed coronavirus disease (COVID-19); therefore, we started a continuous intervention from 69 days after the stroke onset. At 69 days after the stroke onset, the Brunnstrom recovery stage was I, the Fugl–Meyer assessment (FMA) synergy score (FMS) was 0 points, and severe motor paralysis remained. Additionally, the Berg Balance Scale (BBS) score was 24 points, and the trunk impairment scale (TIS) score was 14 points. The functional ambulation category (FAC) was 1. The patient used a wheelchair for daily life in the hospital and hoped to achieve independent walking. Physical therapy was provided for 1 h/day (7 times/week) at this hospital, and assisted walk training was performed using a knee–ankle foot orthosis (Kawamura Gishi Inc., Osaka, Japan).

At 137 days after stroke onset, the patient walked using a T-cane and an ankle–foot orthosis (AFO) with metal uprights and an oil damper at the ankle joint (Gait Solution: Kawamura Gishi Inc., Osaka, Japan). The Brunnstrom recovery stage was II and the FMS was 6 points. The patient’s severe motor paralysis continued. The BBS score was 42 points and the TIS score was 20 points. The FAC was 4. During this period, the participant hoped to walk independently without an AFO. The clinical characteristics of the patient are summarized in Table 1.

This case had no diagnoses of cognitive impairment, psychiatric disorders, or neurological dysfunctions, except for post-stroke sensorimotor deficits. All procedures were approved by the ethics committee of Takarazuka Rehabilitation Hospital of Medical Corporation SHOWAKAI (ethics review number: 20211003) and were conducted in accordance with the tenets of the Declaration of Helsinki. The patient consented to participate in the study.

### 2.2. Experimental Design and Procedure

This study had a retrospective AB design with a follow-up period. We conducted phases A and B for 1 week each, with a follow-up period of 2 weeks. The present study comprised conventional walk training in phase A and a follow-up period. In phase B (144–151 days after stroke onset), tDCS (DC-Stimulator Plus, NeuroConn, Germany) was performed during walking (Table 2, Figure A1).

Electrodes (35 cm^2^; 7 × 5 cm) covered by a saline-soaked sponge were used as the anode and cathode, respectively. The anodal electrode was placed to cover FC2 and FC4 based on the international electroencephalogram 10–20 system [22], which corresponds to the right SMA. The cathode electrode was placed above the contralateral supraorbital region. The current intensity was 1.5 mA and the duration of stimulation was 30 min with a 10-s fade-in and fade-out time [23]. The current density was set at 0.04 mA/cm^2^ (1.5 mA/35 cm^2^) to be below the threshold that leads to tissue damage [18]. A conductive gel was applied under the electrodes to reduce the contact impedance. The use of tDCS in this study was simulated in COMETS2 (http://cone.hanyang.ac.kr/BioEST/Kor/Comets.html accessed on 20 November 2021), a MATLAB-based tDCS toolbox (Figure 2) [24].

### 2.3. Clinical Evaluation

The participant’s performance was assessed using the BBS [25], TIS [26], and FAC [27]. The participant was assessed using the FMA to measure the severity of motor paralysis and the FMS was used to determine the FMA motor score [11]. Muscle strength on the paretic side was assessed for knee joint extension.

### 2.4. Evaluation of Walking and Electromyography

The walking speed and stride time variability (STV) were assessed while the patient walked twice on a 10 m walkway with a supplementary 4 m walkway, assisted by a physical therapist nearby to avoid falling. Data from wireless tri-axial accelerometers, wireless electromyography (EMG), and videos were recorded while the patient walked. Wireless tri-axial accelerometers (Gait Judge System: Pacific Supply Inc., Osaka, Japan; sampling rate: 1 kHz) were attached directly above the lateral malleoli on the paretic side. To minimize the mixing of components in different directions in the accelerometer, the initial angle of the vertical axis in the accelerometer was made as consistent as possible in the shank direction [28].

Intramuscular coherence was calculated from the proximal and distal tibialis anterior (TA) muscle on the paretic side using wireless surface electromyography (Gait Judge System: Pacific Supply Inc., Osaka, Japan; sampling rate: 1 kHz) [29]. To minimize the crosstalk between the pairs of electrodes due to activity from overlapping motor unit areas and adjacent muscles, electrodes were placed at a distance of 10 cm from each other [30]. Corresponding areas of the skin were shaved and cleaned with alcohol before electrode placement. All pre-processing procedures of EMG, including electrode placement, were conducted in accordance with the surface EMG for the non-invasive assessment of muscles guidelines (http://www.seniam.org accessed on 20 November 2021).

### 2.5. Data Analysis

Walking speed was measured by a stopwatch when the participant passed the start and end lines of the 10 m walkway using the recorded video data [31]. After removing the acceleration and deceleration phases from the dataset to prevent any confounding effects, 15 gait cycles were included in the analysis. STV represents the coefficient of variation of stride-to-stride time during walking and is used as an indicator of gait variability [32,33]. The values of STV were calculated by identifying the heel strike timing on the paretic side, using a triaxial accelerometer, and obtaining the mean and standard deviation values of consecutive gait cycles as follows:(1)$$\mathrm{STV}=\frac{\mathrm{Standard}\;\mathrm{deviation}\;\mathrm{value}}{\mathrm{Mean}\;\mathrm{value}}\times 100\;$$

Raw EMG signals were band-pass filtered using a zero-lag 4th-order Butterworth filter with cutoff frequencies of 5–450 Hz, after which they were subtracted as the mean, full-wave rectified. Intramuscular coherence analysis was performed on two time-series EMG signals recorded from the proximal and distal parts of the paretic TA. Coherence analysis was used as a measure of the linear correlation between two EMG signals in each frequency domain. EMG–EMG coherence analysis was performed on full-wave rectified data, which is supposed to increase test-to-test reproducibility and reliability. Coherence can range from 0 to 1, with 1 representing a perfect linear correlation. As the coherence of the beta band (15–30 Hz) was strongly reflected in the corticospinal tract activity, we calculated the beta band mean value in each gait cycle [34,35]. Data at heel strike during walking, detected by a triaxial accelerometer attached above the paretic lateral malleoli, were used to trigger the analysis window of the EMG [36]. The analysis window consisted of 300 ms of data segments extracted from each cycle before and after the heel strike [37]. After the selection of the EMG window, the data were passed through the Hanning window (window length 0.3 s, overlap 0.15 s) and then concatenated [29,38]. We defined the coherence between two concatenated EMG signals (x and y) as the square of the cross-spectrum normalized with the auto-spectrum as follows:(2)$$C_{\mathrm{xy}}\left(f\right)=\frac{|P_{\mathrm{xy}}\left(f\right){|}^{2}}{P_{\mathrm{xx}}\left(f\right)P_{\mathrm{yy}}\left(f\right)}$$
where C_xy_ denotes the amplitude squared coherence for a given frequency (f). P_xx_(f) and P_yy_(f) indicate the x and y power spectra, respectively, and P_xy_(f) is the value of the cross spectrum. The changes in intramuscular coherence during phase A, phase B, and the follow-up period were calculated from the coherence values at all five time points by removing the trend of the slope. MATLAB R2019b (MathWorks, Inc., Natick, MA, USA) was used for all data analyses.

## 3. Results

With the use of tDCS, this case did not present any adverse effects during or after the experiment.

### 3.1. Clinical Evaluations

Table 1 shows the time course of the changes in clinical evaluations. FMS, TIS, and BBS scores did not change between phases A and B. The knee extensor strength on the paretic side was 0.1 kgf higher in phase B than in phase A. The change in walking speed was 0.03 m/s in both phase A and phase B. The change in STV was 0.66% in phase A and −2.23% in phase B.

FMS and BBS scores were higher during follow-up than in phase B. There was no change in the TIS score. The knee extensor strength on the paretic side was 0.3 kgf higher in the follow-up period than in phase B. The change in walking speed was 0.04 m/s and the change in STV was −0.30% in the follow-up period.

### 3.2. EMG Response and TA–TA Coherence in Early Stance and Late Swing

The time series data of the muscle activity and the coherence values of the TA during walking are summarized in Figure 3. EMG response results showed increased muscle activity in the TA distal in the end of phase B. In addition, an increase in TA muscle activity during the early stance phase was seen at the follow-up (arrows in Figure 3). This result suggests that the SMA manipulation increased muscle activity in the TA under the control of the CST. The TA muscle activity seen during the early stance at the follow-up resembled healthy walking, indicating improvement.

### 3.3. TA–TA Coherence in Different Sessions A/B and Follow-Up

Figure 4 shows the time course of the TA–TA coherence changes during the late swing phase. There were no noticeable changes between phases A/B and the follow-up period. In particular, while TA–TA coherence increased at the start of phase B (0.66 (×10^−3^)) compared with that of phase A (0.36 (×10^−3^)), it decreased at the end of phase B (−0.3 (×10^−3^)) and follow-up ((−0.9 (×10^−3^)); Figure 4B). This result suggests that manipulation of SMA excitability has little effect on CST excitability derived from the TA–TA during the late swing phase.

Figure 5 shows the time course of the changes in TA–TA coherence during the early stance phase. While TA–TA coherence noticeably increased in phase B (0.009 and 0.024) compared with phase A (−0.005), it decreased in the follow-up period (−0.044) when compared to phase B (Figure 5B). This result indicates that manipulation of SMA excitability affects CST excitability derived from the TA–TA during the early stance phase.

### 3.4. Comparison of TA–TA Coherence in Early Stance and Late Swing at the A/B and Follow-Up Sessions

Figure 6 shows the comparison of TA–TA coherence during early stance and late swing at the A/B and follow-up sessions. In the start and end of phase A, higher coherence values were seen in late swing (0.0055 and 0.0051) as opposed to early stance (0.0023 and 0.0013). In contrast, these were higher at the end of phase B in early stance (0.043) than late swing (0.0039). These results suggest that a stroke patient with increasingly reduced TA–TA coherence in early stance presented increased CST excitability in that phase resulting from tDCS on SMA. Increased CST excitability in early stance during Phase B improved walking speed and STV, while the FMS and Berg balance scale remained unvaried (Table 1, Figure 5 and Figure 6).

## 4. Discussion

In this longitudinal single-case study, we examined the effects of tDCS in the SMA combined with walk training on CST excitability in a patient with severe motor paralysis of the lower limb. Our results showed that tDCS increasingly affected TA–TA coherence more strongly in the early stance phase compared to the late swing phase. Furthermore, an effect of tDCS was not found during the follow-up, regardless of the gait cycle. The tDCS did not show a significant change in walking speed; however, the STV was noticeably improved. The results of the present study clarified that tDCS over the SMA during walking improved gait stability and enhanced CST excitability in the early stance phase.

MEP recorded from the TA during walking is enhanced in the early stance and swing phases, indicating increased excitability of the CST during these timings [4]. In the present study, the CST was severely damaged by lesions involving the left internal capsule. Although the patient was 137 days post-stroke onset, voluntary movement of the lower limb was severely limited due to significant damage to the CST, which necessitated the use of AFO in daily life. Additionally, the intramuscular coherence of TA–TA during walking decreased over the entire gait cycle—especially in the early stance phase. Therefore, interpreting the results of the patient’s lesion, physical function, and walking assessment, the excitability of the CST was considerably reduced.

Interestingly, TA–TA intramuscular coherence improved only in the early stance phase when tDCS over the SMA was combined with walk training in the patient. This result indicates that the increase in excitability of the SMA using tDCS had a greater impact in the early stance phase than during the swing phase. In previous studies, outcomes in post-stroke patients with a damaged CST depended on the recovery of the CST via the SMA [13]. Moreover, although the CST generates TA muscle activity, it may be triggered through the SMA due to damage to the CST [39]. This is also possible due to the influence of the corticoreticular tract originating from the SMA [39]. Therefore, we consider that the excitability of the CST and the corticoreticular tract originating from the SMA is enhanced using tDCS, resulting in increased TA–TA coherence during the early stance phase. On the other hand, TA–TA coherence during the swing phase may require CST activity originating from M1. This result supports the stronger involvement of the CST during the swing phase than during the stance phase [40].

Regarding walking performance, tDCS over the SMA affected the STV, not walking speed. STV indicates the variability in stride-to-stride time, reflecting rhythmic walking [41]. Additionally, STV is strongly associated with balance function and falls [32,42]. Therefore, we consider STV to be a marker of gait stability. MEP in the early stance phase derived from the TA during walking is considered to be a postural stability strategy for heel strike impact [5]. Additionally, because SMA is related to balance ability [43], increased SMA excitability may improve dynamic balance during walking. Therefore, the increase in TA–TA coherence during the early stance phase is considered to have enhanced the reaction after heel strike and then reduced STV.

This study had some limitations. First, AFO was used during the gait measurements in this study. The walking measurement used an AFO. AFO affects the movement of the lower limbs during the swing and early stance phases. However, AFO allowed 15° of dorsiflexion/20° of plantar flexion of the ankle joint in this study, which suggests that the influence of AFO is minimal. Second, it is possible that phase B did not have a carry-over effect to follow-up because of the short intervention of seven sessions/week. The present study is within this range as previous studies have conducted 6–12 sessions during walking [44]. Nevertheless, these numbers of sessions are difficult to compare simply because the endpoints differ in walking parameters and intramuscular coherence. Finally, because this study is a retrospective single case study, the findings need to be examined in a population using a prospective study design. We consider a further prospective study to be needed to clarify the role of tDCS by comparing active versus sham stimulation of tDCS, and the net effect of physical therapy combined with tDCS and physical therapy only. The present study showed that tDCS over the SMA combined with walk training increased TA–TA intramuscular coherence during the early stance phase and reduced STV. These results partly explain the functional implications and concept of walk training for the improvement of CST excitability during early stance. Therefore, this single-case study is useful for the neuromodulation of walking because the results reveal the clinical application of tDCS for this specific type of post-stroke recovery.

## 5. Conclusions

The tDCS over the SMA combined with walk training enhanced the TA–TA intramuscular coherence of the beta band in the early stance phase. This tDCS intervention also reduced STV as an indicator of temporal gait variability. These results suggest that an increase in the TA–TA intramuscular coherence of the beta band in the early stance phase possibly enhances the response after heel strike, resulting in the reduction of STV. Further investigation of this hypothesis is warranted in future studies.

## Figures and Tables

**Figure 1 brainsci-12-00540-f001:**
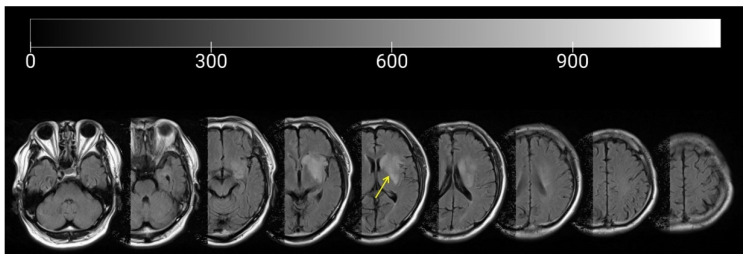
Magnetic Resonance Imaging of the patient. This figure shows the magnetic resonance imaging of the patient. Yellow arrow indicates the lesion site. Values at the top of the image indicate the level of brightness, with values closest to 0 representing increasing darkness and values closest to 900 indicating increasing brightness.

**Figure 2 brainsci-12-00540-f002:**
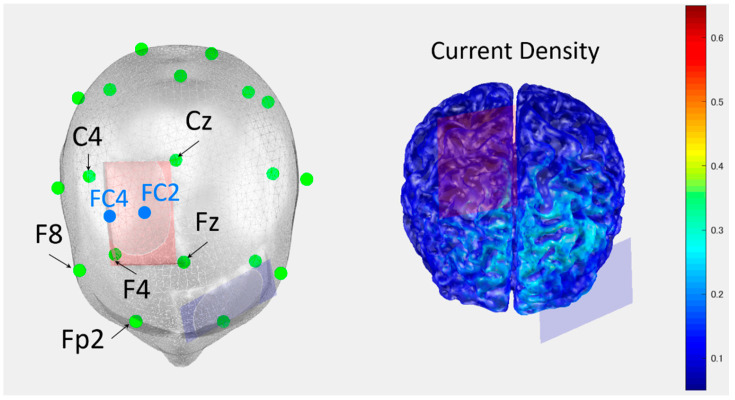
The simulation of tDCS over the SMA on the damaged side. This figure shows the simulation of tDCS over the SMA on the damaged side. This value indicates the current density (J). Cz indicates the midpoints of the distance between the nasion and Inion, and between the left and right preauricular points. Fz indicates the position of 20% of the distance between nasion and inion in the nasion direction. C4 indicates the position of 20% of the distance between the preauricular points in the direction of the right preauricular. F4 indicates the midpoint between Fz and F8, and between C4 and Fp2. FC indicates the midpoint of F and C corresponding to each number.

**Figure 3 brainsci-12-00540-f003:**
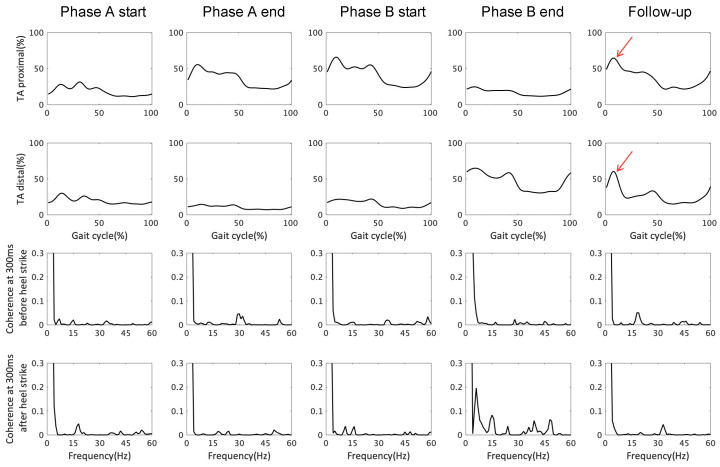
Time series data of tibialis anterior (TA) muscle activity and coherence values during walking. From left to right, phase A starts and ends, phase B starts and ends, and the follow-up are shown. From the top, the vertical axis shows TA muscle activity (proximal and distal), coherence at 300 ms before heel strike (i.e., late swing), and coherence at 300 ms after heel strike (i.e., early stance). The horizontal axis indicates the gait cycle (%) and frequency (Hz). Red arrows indicate increased TA muscle activities during the early stance phase at the follow-up. Note that the TA muscle activities (%) were band-pass filtered using a zero-lag 4th-order Butterworth filter with cutoff frequencies of 20–450 Hz, demeaned, rectified, and low-pass filtered using a zero-lag 4th-order Butterworth filter with a cutoff frequency of 10 Hz. Abbreviations: TA proximal, proximal part of tibialis anterior; TA distal, distal part of tibialis anterior; Coherence, intramuscular coherence between the proximal and distal part of the tibialis anterior.

**Figure 4 brainsci-12-00540-f004:**
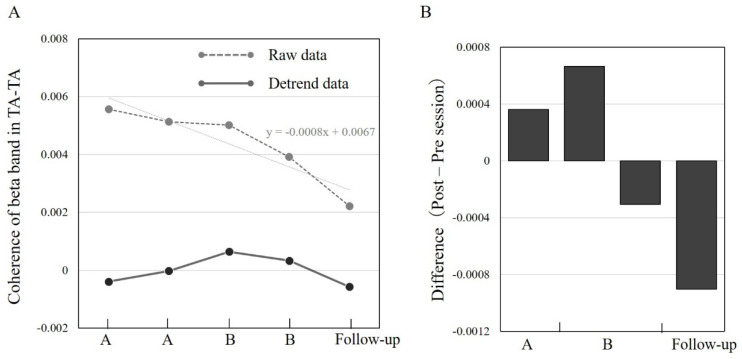
Coherence values extracted from the TA–TA before 300 ms of paretic heel contact during walking. The study’s experimental design was an AB design with a follow-up. Phases A and B were set at 1 week each, and the follow-up was conducted at 2 weeks. (**A**) shows the coherence of the beta band in TA–TA. The columns indicate, from left to right, the start and end of phase A, the start and end of phase B, and the follow-up. Detrend data was used to remove the influence of natural recovery from between phases. (**B**) shows the mean difference in the coherence of the detrend data between the pre-and post-session (Phase A/B and follow-up). Note that phase B has two results for different phases (Left bar: End of phase A and start of phase B, right bar: start of phase B and end of phase B). Abbreviation: TA, tibialis anterior.

**Figure 5 brainsci-12-00540-f005:**
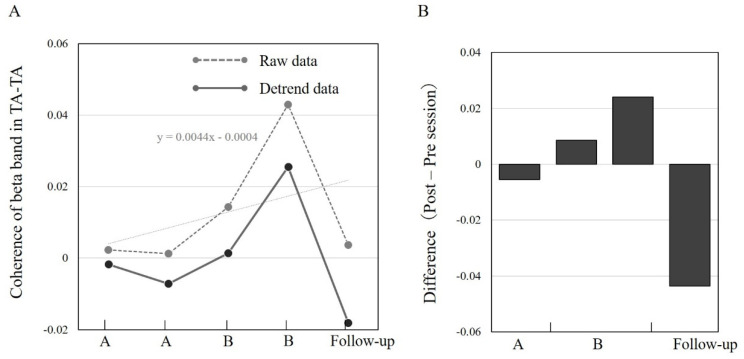
Coherence values extracted from the TA–TA after 300 ms of paretic heel contact during walking. The study’s experimental design was an AB design with a follow-up. Phases A and B were set at 1 week each, and the follow-up was conducted at 2 weeks. (**A**) shows the coherence of the beta band in the TA–TA. The columns indicate, from left to right, the start and end of phase A, the start and end of phase B, and the follow-up. Detrend data was used to remove the influence of natural recovery from between phases. (**B**) shows the mean difference in coherence of detrend data between the pre-and post-session (Phase A/B and follow-up). Note that phase B has two results for different phases (Left bar: End of phase A and start of phase B, right bar: start of phase B and end of phase B). Abbreviation: TA, tibialis anterior.

**Figure 6 brainsci-12-00540-f006:**
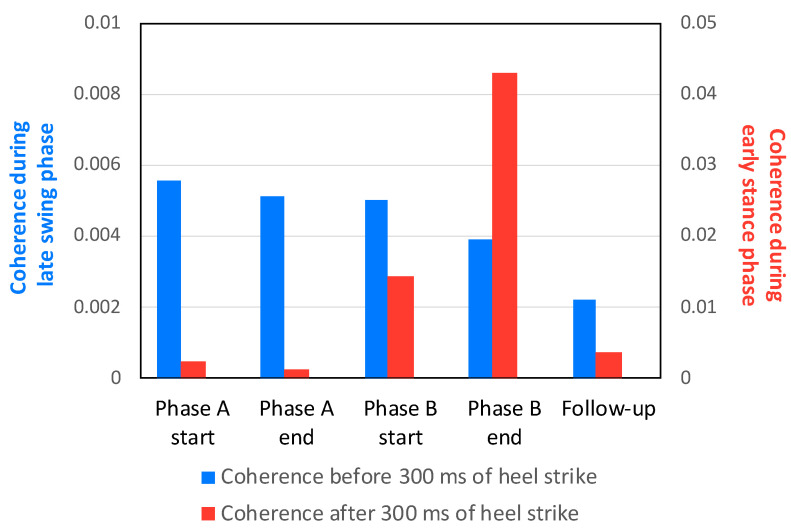
Comparison of TA–TA coherence values in early stance and late swing at the A/B and follow-up sessions. The blue bars indicate TA–TA coherence during the late swing phase, while that during the early stance phase is represented by red bars. Note the differing labels for the vertical axes between colored bars.

**Table 1 brainsci-12-00540-t001:** Patient’s clinical characteristics.

		Stroke Onset (Days)				
		69	137	144	151	165
Functional Ambulation Category		1	4	4	4	4
BRS (lower extremity)		1	2	2	2	2
FMS (lower extremity): max = 22 *	Total score	0	6	6	6	9
	Flexor synergy	0	0	0	0	2
	Extensor synergy	0	5	5	5	5
	Knee Ankle when sitting	0	1	1	1	2
	Knee Ankle when standing	0	0	0	0	0
FMA sensory score (lower extremity): max = 12 †		6	8	8	8	8
Knee extensor strength (kgf)		0	2.4	2.4	2.5	2.8
Trunk Impairment Scale		14	20	20	20	20
Berg Balance Scale (scores)		24	42	42	42	46
Comfortable walking speed (m/s)		-	0.45	0.48	0.51	0.55
Stride Time Variability (%)		-	4.00	4.66	2.43	2.13

* Synergy score of Fugl–Meyer assessment. † Sensory score of the Fugl–Meyer assessment. Note that 137 days, 144 days, and 151 days after stroke onset correspond to the start of phase A, phase B, and a follow-up period. Interventions with tDCS (phase B) were performed from 144 to 151 days. Abbreviations: BRS, Brunnstrom Recovery Stage; FMS, Fugl–Meyer synergy score; FMA, Fugl–Meyer assessment.

**Table 2 brainsci-12-00540-t002:** Experimental protocol overview.

Items		Details
Timeseries information		
Timeline of stroke		Functional Ambulation Category improved from 1 to 4 between 69 and 137 days from stroke onset Phase A: 1 week from 137 days after stroke onset, Phase B: 1 week from 144 days after stroke onset, Follow-up: 2 weeks from 151 days after stroke onset
Instrument settings		
tDCS stimulation	Placement	Anode: supplementary motor cortex on the damaged side, cathode: contrala-teral supraorbital region
	Parameter setting	Current intensity: 1.5 mA, duration of stimulation: 30 min (Phase B)
Data collecting		
Measurements	Clinical evaluation	Brunnstrom Recovery Stage and Synergy score of the Fugl–Meyer assessment, Knee extensor strength (Motor paralysis) Sensory score of the Fugl–Meyer assessment, Trunk Impairment Scale, Berg Balance Scale
	Walking assessment	Comfortable walking speed, Stride time variability (Gait stability)
	EMG assessment	TA-TA coherence in beta band (Corticospinal tract excitability) during late swing and early stance phases

## Data Availability

The dataset presented in this study is available upon request from the corresponding author. Data cannot be shared publicity because the participant did not consent for public sharing.

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
