# Peer review of "Effects of Transcranial Direct Current Stimulation over the Supplementary Motor Area Combined with Walking on the Intramuscular Coherence of the Tibialis Anterior in a Subacute Post-Stroke Patient: A Single-Case Study"

_brainsci, 2022, doi:10.3390/brainsci12050540_

Round 1
Reviewer 1 Report
This manuscript presents the case report findings in a clear and straight-forward manner. The only minor comment of this reviewer is considered a just suggestion left to the authors discretion.
Lines 18 & Line 248 While it is acceptable to use the term “walking training”, I suggest using the term “walk training” to avoid adjacent words ending in “ing” which reads somewhat awkwardly. I believe the words "walking training" only occurs twice within the manuscript.
Reviewer 2 Report
The study exploring the impact of tDCS of SMA on walking parameters post stroke is very interesting. However I think the data present is not yet sufficient for publication, and that the case study approach is not sufficient given the tDCS intervention is novel, without a sham control, and not repeated in a second training phase on the patient.
Major concerns:
Can the authors detail why no sham stimulation was used in Phase A?
A second phase including tDCS after Phase B would have provided more evidence for the suggested effects of tDCS. Why was a repeat application of tDCS with training not considered?
Minor corrections:
L117: reads "the anodal was placed", but should read "The anode was placed" or "the anodal electrode was placed"
Figure 2: There are no labels of FC2 and FC4 on the montage diagram, but there is a label of FZ. Labelling of FC2 and FC4 should occur, particularly as the figure implies other 10-20 points.
Further the authors should clarify how the positioning of the electrodes was determined - was this by measurement in reference to CZ (hence the inclusion of CZ) or by some other method?
The authors should detail the rationale behind the selection of the specific electrode sizes, and also why they chose to have two electrodes that were the same size rather than choosing alternative montages.
L142: made as consistent as possible in the ....."
L186: what adverse effects were considered, was the participant given a questionnaire?
Reviewer 3 Report
Reviews
- Abstract
- The abstract should discuss a little more about the score you have used and how you have quantified stability. Thus, it can be validated that stability has improved due to the excitability of the SMA region.
- Introduction
- Line 45-47, Will you be more explicit about how the contralateral side is controlled effectively due to certain physiological adaption of the SMA region. The nerve fibers increased from the SMA region based on the study you cited. It will be nice if you will be clear on that.
- Methods
- For figure 1 kindly provide the information regarding the scale at the top of the image. 0, 300, 600, 900. For readers not familiar with MRI, it will give them an idea about the depth of slices and the numeric values and units associated with it.
- I recommend adding a table showing the timeline of stroke, the introduction of stimulation, and other measures used in a study (A/B design). It will be easier to understand the experimental protocol.
- Results
- Figure 4 needs to be more clearly defined. For example, what does each bar graph define? I cannot understand what the xtick labels define for both the line plot and bar graphs. I can see one bar under A and two bar graphs under B and one bar graph under follow-up. Will you kindly be clearer in defining graphs or presenting them?
- Also, what is the point of detrending the data? Such kind of statistical transformation skews the data. I am not able to understand why you are comparing your transformed data (processed) with pre-transformed (pre-processed) data. What is the rationale behind it? It is quite clear from the slope that the changes in coherence over the sessions are decreasing and are way steeper towards the follow-up session.
- Why did the coherence increased in B and then decreased during follow upsession.
- I also suggest you be clear about what do these results mean in terms of rehabilitation. Do these results validate your hypothesis that the excitability of SMA improves walking which is reflected via lower coherence of TA muscles? Try to break your results into more subsections, with the graph for each subsection explaining the story of your study. I encourage a narrative of your results validating the claim of your study in the text of these subsections or as a caption for the figure. For example, I will write a subsection about “Excitability of SMA is higher in stance than swing phase”, support it with a figure or graph, and then at the end of the text, I will use a narrative to explain these results and their association with improved rehabilitation. This will improve the quality of your work and results.
- I suggest adding another graph where the difference in coherence for early stance and late swing is clear displayed or shown (either a bar plot or something else).
- Overall, I think the result section needs to be improved significantly and more organization is needed for it.
- Discussion
- The stability of gait can be quantified using some measure such as angular momentum, the base of support. I would recommend you to be a little explicit about the measure you are using to quantify gait stability. In your case, the measure is different scoring assessments. If you will be clear about it, it will be great.
- The reduction in excitability of CST is reflected through reduced TA-TA coherence in the swing phase or the increase in excitability of CST which is reflected via increased TA-TA coherence in the stance phase. How can you determine whether one is associated with improved walking control or not as both stance and swing are necessary for a smooth gait transition? Effective inhibition of neurons in spinal circuitry can result in the reduction of excitability during the stance or swing for a stable gait.
- Moreover, if you are suggesting it is occuring over the course of time (increased coherence in early stance) and can be associated with imporved control then you should have control to test it. The effect of motor learning can also be associated with this reduced or increased TA-TA coherence rather than just induced excitability of the SMA region. How can you defend this question in your study?
- Can you please also show increment in stance coherence over B and follow up sessions. Why did you see an increase and decreased in coherence in followup session. Can you explain a neural mechanism behind it more clearly.
- Overall, I like the way the discussion is written.
- Conclusion
- The conclusion should be a separate section.
Overall, I find the study interesting. It is performed nicely. The major concern I can bring about the study is the organization of the results. The results need to be more organized and should be broken down into more subsections. I would break it down as suggested below. However, this is a suggestion authors can use their ways to do it. The graphs and figure font size should be increased so that they can be easily read.
- EMG response in early stance and late swing.
Figure, Result, and simpler narrative of what these results mean from a neurophysiological perspective.
- EMG TA-TA coherence in different sessions A/B/Follow-up
Figure, Result, and simpler narrative of what these results mean from a neurophysiological perspective.
- Comparison of EMG coherence in early stance and swing for A/B and Follow-up sessions
Figure, Result, and simpler narrative of what these results mean from a neurophysiological perspective.

Round 2
Reviewer 3 Report
Kindly present images that are not snapshots of matlab. Based on the small timeline for the second review. I briefly wen through the manuscript and can suggest one more minor revision.
Figure 6 should also show standard deviation for the bar graph as error bars. This should be applied to other quantitative data.
Moreover, in the experimental design figure. Can you be more clear that its 1 week from stroke rather than just writing 1 week.
